# Magnetically Actuated Microscaffold with Controllable Magnetization and Morphology for Regeneration of Osteochondral Tissue

**DOI:** 10.3390/mi14020434

**Published:** 2023-02-11

**Authors:** Junhyeok Lee, Hyeong-Woo Song, Kim Tien Nguyen, Seokjae Kim, Minghui Nan, Jong-Oh Park, Gwangjun Go, Eunpyo Choi

**Affiliations:** 1School of Mechanical Engineering, Chonnam National University, 77 Yongbong-ro, Buk-gu, Gwangju 61186, Republic of Korea; 2Robot Research Initiative, Chonnam National University, 77 Yongbong-ro, Buk-gu, Gwangju 61186, Republic of Korea; 3Korea Institute of Medical Microrobotics, 43-26, Cheomdangwagi-ro 208-beon-gil, Buk-gu, Gwangju 61011, Republic of Korea

**Keywords:** microscaffold, magnetic actuation, tissue regeneration

## Abstract

Magnetic microscaffolds capable of targeted cell delivery have been developed for tissue regeneration. However, the microscaffolds developed so far with similar morphologies have limitations for applications to osteochondral disease, which requires simultaneous treatment of the cartilage and subchondral bone. This study proposes magnetically actuated microscaffolds tailored to the cartilage and subchondral bone for osteochondral tissue regeneration, named magnetically actuated microscaffolds for cartilage regeneration (MAM-CR) and for subchondral bone regeneration (MAM-SBR). The morphologies of the microscaffolds were controlled using a double emulsion and microfluidic flow. In addition, due to their different sizes, MAM-CR and MAM-SBR have different magnetizations because of the different amounts of magnetic nanoparticles attached to their surfaces. In terms of biocompatibility, both microscaffolds were shown to grow cells without toxicity as potential cell carriers. In magnetic actuation tests of the microscaffolds, the relatively larger MAM-SBR moved faster than the MAM-CR under the same magnetic field strength. In a feasibility test, the magnetic targeting of the microscaffolds in 3D knee cartilage phantoms showed that the MAM-SBR and MAM-CR were sequentially moved to the target sites. Thus, the proposed magnetically actuated microscaffolds provide noninvasive treatment for osteochondral tissue disease.

## 1. Introduction

Knee joint trauma and bone joint friction injuries occur in humans regardless of their age. In particular, osteoarthritis, which occurs frequently in the elderly, is one of the most common diseases of the global aging society [1,2]. Among the various treatment methods used in knee joint cartilage regeneration, cell-mediated therapy is aimed at relieving symptoms and regenerating damaged osteochondral tissues through direct cell injections and scaffold transplantations [3,4,5]. Direct cell injection is a minimally invasive procedure, but the number of cells delivered to the lesion is low due to the absence of active targeting [6,7]. On the other hand, cell-loaded scaffold transplants can be safely delivered to lesion sites by loading cells onto biocompatible and biodegradable polymer-based scaffolds [8,9]. However, the sizes of the scaffolds commercialized to date are on the centimeter scale; thus, scaffold transplantations require invasive surgery involving knee incision.

Recently, scaffolds that are capable of cell loading and with a micrometer size and wireless actuation have been developed to overcome the limitations of cell-mediated therapies [10,11,12]. Spheroid microrobots have also been developed, which are made by co-culturing cells and magnetic nanoparticles (MNPs) [13,14]. Among these, wirelessly actuated microscaffolds can be loaded with stem cells and have demonstrated differentiation into chondrocytes [10]. Wireless targeting of a microscaffold using an external magnetic field was demonstrated in the knee joint phantom and ex vivo porcine knee cartilage. Furthermore, these microscaffolds delivered cells for knee cartilage defects in rabbits in vivo to effectively regenerate damaged cartilage. However, knee cartilage damage occurs not only in the cartilage but also in the subchondral bone. Therefore, combined regeneration of hierarchical osteochondral tissue using microscaffolds remains challenging due to the separation between the cartilage and the subchondral bone or the difficulty of spatial control over the differentiation of the transplanted stem cells [15]. Therefore, research on wirelessly actuated scaffolds for combined regeneration of these damaged osteochondral tissues is needed.

This study proposes magnetically actuated microscaffolds capable of osteochondral regeneration in knee joints through a one-step targeted delivery. Magnetically actuated microscaffolds are classified into the following two types: for cartilage regeneration (MAM-CR) and for subchondral bone regeneration (MAM-SBR) (Figure 1a). These microscaffolds consist of three-dimensional (3D) poly(lactic-co-glycolic acid) (PLGA) microstructures for cell loading and surfaces to which MNPs are attached for magnetic targeting. The microscaffolds have a size of hundreds of micrometers, which is much smaller than the existing commercially available cm-scale scaffolds. For the elderly who are physically burdened by surgical incisions, the microscaffolds can be implanted into osteochondral defects through minimally invasive surgery. Additionally, they have pore sizes [3,16] suitable for regenerating cartilage and subchondral bone, along with different sizes [3,16] to realize a one-step targeted delivery based on the relative speed differences between the different magnetizations. As shown in Figure 1b, the MAM-CR and MAM-SBR are loaded with cells and injected into the joint cavity. Then, under the same magnetic field strength, the MAM-SBR, which is relatively larger in size than the MAM-CR, reaches the damaged subchondral bone first at a high speed due to the high magnetic moment caused by the greater number of attached MNPs. The relatively slow MAM-CR is then delivered to the damaged cartilage. The osteochondral treatment method using the differences in speed between these two microscaffolds can assist in the combined regeneration of hierarchical osteochondral tissues by preventing separation between the subchondral bone and providing space for the proliferation and survival of the transplanted cells. Additionally, the microscaffold-based osteochondral minimally invasive procedure can treat bone and cartilage simultaneously, reducing the number of separate surgeries.

## 2. Materials and Methods

### 2.1. Materials

For the fabrication of the PLGA microcarrier, PLGA (with a molecular weight of about 66,000–107,000), dichloromethane (DCM), Span^®®^ 80, PVA (with a molecular weight of ~61,000), gelatin (from porcine skin; gel strength 300; Type A), dopamine hydrochloride, Trizma^®®^ hydrochloride solution (pH 8.0; BioPerformance Certified; 1 M; suitable for cell culture; from Sigma-Aldrich), and iron(II,III) oxide (nanopowder; with a particle size of 50–100 nm (SEM); 97% trace metal basis) were purchased from Sigma-Aldrich (St. Louis, MO, USA). For the PDA coating on the microscaffolds, dopamine hydrochloride was purchased from Sigma-Aldrich.

### 2.2. Fabrication of the Magnetic Microscaffolds

The magnetic microscaffolds were fabricated by w-o-w double emulsion and MNP attachment. The fabrication method was modified from our previous work [10]. The PLGA solution was prepared by dissolving PLGA (80 mg/mL for the bone and 72 mg/mL for the cartilage) and 15 µL of Span 80 in DCM. The PLGA solution was mixed with the gelatin aqueous solution (7.5 *w*/*v*% for the bone and 10 *w*/*v*% for the cartilage) prepared in 1% PVA using a vortex mixer at a ratio of 5.5:4.5. The prepared water-in-oil (w-o) emulsion was placed in a 3 mL syringe and mounted on a syringe pump; then, the 1% PVA solution was placed in a 20 mL syringe and mounted on a second syringe pump. For the microscaffold for subchondral bone regeneration, the w-o emulsion and the 1% PVA solutions were connected to 22 and 17 gauge needles to form w-o-w emulsion beads at flow rates of 0.05 mL/min and 5 mL/min, respectively. In addition, the solutions for the microscaffold for cartilage regeneration were connected to 26 and 21 gauge needles to form beads at flow rates of 0.05 mL/min and 3 mL/min, respectively. The w-o-w emulsion microbeads were solidified by evaporation of the DCM in deionized water at 4 °C. Then, the gelatin in the microbeads was removed using deionized water at 40 °C, and the PLGA microscaffolds were obtained. An aqueous solution of dopamine hydrochloride was dissolved in a 10 mM Trizma^®®^ hydrochloride solution at a concentration of 2 mg/mL, and the prepared microscaffolds were coated in the dopamine hydrochloride for 90 min. The microscaffolds coated with the PDA were washed with the 10 mM Trizma^®®^ hydrochloride solution and placed in an 80 mg/mL MNP aqueous solution for 3 h to attach the MNPs. The magnetic microscaffolds were then washed with deionized water three times and stored in deionized water until use.

### 2.3. Characterization of Magnetic Microscaffolds

The morphologies of the microscaffolds were determined by SEM (SU8010; HITACHI, Japan). The elemental signals of the microrobots were detected by EDX for semi-quantitative analysis. The diameters and pore sizes of the microscaffolds (number of MAM-CR = 43 and number of MAM-SBR = 69) were analyzed using the ImageJ software (National Institutes of Health, Bethesda, MD, USA). The magnetization curves of the microrobots and their components were recorded using a vibration sample magnetometer (VSM) (Lake Shore Cryotronics 7404, Westerville, OH, USA).

### 2.4. Cell Culture

Murine bone marrow-derived D1 mouse mesenchymal stem cells and mouse C2C12 myoblast cells were cultured in Dulbecco’s modified Eagle’s medium (DMEM; Sigma-Aldrich, MO, USA) containing 10% (*v*/*v*) fetal bovine serum (FBS; Sigma-Aldrich, MO, USA), 1% (*v*/*v*) penicillin–streptomycin (PS; Sigma-Aldrich, MO, USA), and 1% (*v*/*v*) amphotericin (Sigma) at 37 °C in humidified 5% CO_2_ conditions. The medium was replaced every 2–3 days, and passage 5 cells were used in all of the experiments.

### 2.5. Cell Viability

To confirm the viability of the D1 and C2C12 cells on the MAM-CR and MAM-SBR for osteogenesis and chondrogenesis, the MAM-CRs and MAM-SBRs were sterilized with 70% ethanol for 30 min and washed twice with PBS. Then, each type of cell was seeded onto the MAM-CR and MAM-SBR and placed on ultralow attachment surface 96-well plates (Corning, NY, USA) at a density of 5000 cells/microscaffold. The Alamar Blue cell viability reagent (Invitro, OR, USA) was used for the viability tests on days 1 and 3.

To observe the cells that were attached to the microscaffolds, the cells were stained with CellTrace (ThermoFisher, MA, USA) and then cultured at 5000 cells per microrobot in ultralow attachment round bottom 96-well plates for 24 h. Thereafter, the cell-loaded microscaffolds were fixed using 4% paraformaldehyde, and the nucleoli of the cells inside the microscaffolds were stained with 4′,6-diamidino-2-phenylindole (DAPI). Lastly, the microscaffolds with the cells were observed under a confocal microscope (LSM 880; Carl Zeiss, Oberkochen, Germany).

### 2.6. Magnetic Actuation of Microscaffold

In this study, the microscaffolds were driven magnetically through pulling motions using a gradient magnetic field generated from an electromagnetic actuation (EMA) system and a permanent magnet. Under the magnetic field and gradient, the microscaffolds experience a magnetic torque (**τ**) and magnetic force ($F_{m}$) as follows:(1)τ=VM×B
(2)$$F_{m}=V\left(M\cdot \nabla \right)B$$

The volume of the microscaffold, the magnetization vector of the microscaffold, the magnetic flux density, and the gradient symbol are represented by *V*, $M\in R^{3\times 1}$, $B\in R^{3\times 1}$, and **∇**, respectively.

For the locomotion tests of the magnetic microscaffolds, nine electromagnetic coils and pure iron cores for the EMA system were provided by JL Magnet (Republic of Korea). After fabrication of the EMA system, a Gauss meter (8030, FW Bell, Milwaukie, OR, USA) was used to measure the magnetic field generated by the coils for comparison with the simulation results. An optical microscope (F170; Carl Zeiss, Berlin, Germany) and a DSLR camera (EOS 600D; Canon, Tokyo, Japan) were used to observe the locomotion of the microrobots. The EMA system was controlled using the LabVIEW software (National Instruments, Austin, TX, USA), and the current in each coil was applied through six power supplies (each set of MX 15 and 3001 ix; AMETEK, Berwyn, PA, USA).

To perform targeting tests of the magnetic microscaffolds, a cube-shaped acrylic chamber was prepared by laser cutting and glue attachment. The chamber was filled with a glycerol solution to mimic the viscosity of synovial fluid. Next, a knee joint phantom was fabricated using a 3D printer (Objet30 Pro; Staratsys Ltd., Eden Prairie, MN, USA). The bottom of the phantom was blocked with a slide glass to observe the magnetic targeting of the microscaffolds.

### 2.7. Statistics and Data Analysis

Comparisons of the experimental data were performed using the Student’s *t*-test. All data were presented as means ± standard deviations (SDs). Differences between groups with * *p* values of <0.05 were considered statistically significant.

## 3. Results and Discussion

### 3.1. Characterization of Magnetic Microscaffold

In this study, the magnetic microscaffolds are categorized into two types, for cartilage and for bone regeneration, and have different sizes and pore sizes. They are fabricated sequentially through a water-in-oil-in-water (w-o-w) emulsion, microfluidic flow control, and MNP attachment. As shown in the fabrication diagram of Figure 2a, the PLGA–gelatin emulsion used as a water-in-oil emulsion is injected with a 1% polyvinyl alcohol (PVA) solution inside a microfluidic channel to form spherical microbeads using the w-o-w double-emulsion method. Here, to obtain the MAM-CR and MAM-SBR with different sizes and pore sizes, we adjusted the flow speed of the PLGA–gelatin emulsion in the microfluidic channel. The PLGA porous microscaffold is obtained by removing the gelatin beads in the PLGA–gelatin microscaffold using warm deionized water (DW). Scanning electron microscopy (SEM) images show the morphologies of the fabricated microscaffolds (Figure 2b,c). Based on the SEM images, the sizes and pore sizes of the microscaffolds were analyzed. The MAM-CR had a size of 285.09 ± 17.38 µm and a pore size of 63.82 ± 9.91 µm (Figure 2d), and the MAM-SBR had a size of 771.09 ± 46.40 µm and a pore size of 79.16 ± 11.00 µm (Figure 2e).

To impart magnetic actuation function to the microscaffolds, MNPs were attached to the polydopamine (PDA)-coated PLGA microscaffold surfaces (Figure 2a). Here, the PDA was formed through the polymerization of dopamine; as a mussel adhesive protein, it was coated on the microscaffold surface to attach the MNPs [17]. Surface modifications with the PDA coating can be easily performed with a one-step immersion of the microscaffold in a nontoxic solution [18]. As microscaffolds with attached MNPs, MAM-CR and MAM-SBR were analyzed in terms of their morphologies and compositions through SEM and energy-dispersive X-ray spectrometry (EDX). The SEM images showed that the porous structure was maintained regardless of the attachment of the MNPs (Figure 2f). The EDX mapping images showed that C, O, and Fe were detected on the MAM-CR and MAM-SBR (Figure 2g). Here, C and O are the main elements of the PLGA microscaffold, and Fe is the main element of the MNPs. Furthermore, the energy peak graphs obtained through the EDX analysis also revealed the energy peaks of C, O, and Fe as components of the magnetic microscaffolds (Appendix A). Here, the energy peak for the Pt signal appeared by coating a Pt membrane on the surface of the magnetic microscaffold to reduce the signal noise in the SEM and EDX analyses. Next, the magnetic properties of the MAM-CR and MAM-SBR were investigated through magnetization curves. We measured the magnetization values of 800 magnetic microscaffolds under an external magnetic field using a VSM and calculated the magnetic field response of one microscaffold as a magnetization curve. Before measuring the magnetic responsive behavior of the microscaffolds, we confirmed the magnetization curve of the MNPs that were used for the microscaffold attachment with a magnetic saturation value of about 80 memu/mg (Figure 2h). As a result, the saturated magnetization values of one MAM-CR and one MAM-SBR were 0.058 memu and 0.139 memu, respectively (Figure 2i and Appendix A). Thus, the MAM-SBR has about three times greater saturated magnetization than the MAM-CR. These results implied that the MAM-SBR can move faster than the MAM-CR under a given magnetic field and a gradient magnetic field. In addition, when comparing the magnetic saturation values between the MNPs and the microscaffolds, we estimated that 0.77 μg and 1.84 μg of the MNPs were attached to the MAM-CR and MAM-SBR, respectively.

### 3.2. Cell Loading and Viability Tests of the Magnetic Microscaffolds

To confirm the cell loading and biocompatibilities of the magnetic microscaffolds proposed for osteochondral regeneration, we cultured murine bone marrow-derived D1 mouse mesenchymal stem cells and mouse C2C12 myoblast cells in the MAM-CR and MAM-SBR, respectively. On day 1 of culture, the cells were predominantly located around the magnetic microscaffold. As the culture period increased, the cells adhered harmoniously to the magnetic microscaffolds (Figure 3a,b). In addition, we observed fluorescence images of the cells through confocal microscopy to confirm their viability and adhesion to the magnetic microscaffold. Figure 3c,d shows that the nucleus- and cytoplasm-stained cells were supported on the MAM-CR and MAM-SBR, indicating that they can serve as scaffolds for cell culture. Furthermore, to verify the role of the magnetic microscaffold as a cell carrier, the viabilities of the D1 and C2C12 cells were evaluated with the magnetic microscaffolds for 3 days. For the D1 and C2C12 cells, the MAM-CR and MAM-SBR did not reveal significant differences in cell viability compared to the cell spheroids (Figure 3e,f). These results showed that neither the MAM-CR nor the MAM-SBR have any effect on the cell viability, thereby posing little toxicity to the cells. We used ferromagnetic materials as the MNPs in this experiment, and it is already known that this material does not affect cells [19,20,21,22]. In addition, we confirmed that the cell viability was slightly higher on the microscaffold than on the cell spheroid. This phenomenon happens because spheroids have a high cell density, and the cells inside the spheroids are exposed to an environment that lacks nutrients and oxygen and eventually die [23]. On the other hand, the microscaffold provides a frame to which the cells can attach and grow, allowing sufficient nutrients and oxygen to be supplied to the cells inside the microscaffold. A microscaffold magnetized by an external magnetic field generates strong magnetic fields and a gradient around it [24]. Cells attached to the microscaffold can be affected in terms of cell viability and differentiation by the magnetic field generated from the magnetized microscaffold [25,26,27]. To estimate the magnetic fields generated from the microscaffold, we investigated the magnetic fields and gradients generated from a microscaffold magnetized by an external magnetic field through numerical simulation (COMSOL, COMSOL Inc., MA, USA). As a result, when a magnetic field of 40 mT, which can be generated in the electromagnetic actuation (EMA) system used in this study, is applied to a microscaffold of about 285 µm, magnetic fields of 50.27 mT and 49.13 mT are indicated on the surface of the microscaffold and 10 µm away from the surface, respectively (Appendix A). Through these results, we confirmed that a magnetic field gradient of 114.7 T/m was generated around the magnetized microscaffold. In future work, we will confirm the effect of a microscaffold magnetized by an external magnetic field on cells, both in vitro and in vivo.

### 3.3. Mobility Test of Magnetic Microscaffold

Prior to targeting the MAM-CR and MAM-SBR, we evaluated the mobilities of these magnetic microscaffolds under an external magnetic field generated from an EMA system. The EMA system, composed of nine electromagnetic coils, enables five (two rotational and three translational) degrees of freedom motions of magnetic objects in the workspace. In addition, the EMA system can simultaneously generate a magnetic field of up to 70 mT and a gradient magnetic field of 1.7 T/m [28]. Using the experimental setup and the EMA system (Appendix A), the mobilities of the MAM-CR and MAM-SBR were investigated in phosphate-buffered saline (PBS) under the same magnetic field and several gradient magnetic field ranges. As shown in Figure 4 and Appendix A, the MAM-SBR traveled a longer distance in a given time than the MAM-CR under the same magnetic field strength. This result means that the magnetic force of the MAM-SBR is higher than that of the MAM-CR and similar to the magnetization measurement results (Figure 4 and Appendix A). Furthermore, we measured the speed of the magnetic microscaffolds in PBS under several magnetic fields and a range of gradient magnetic fields. The speed of the magnetic microscaffolds showed a tendency to increase with increases in both the magnetic field and the gradient magnetic field (Figure 4c). In particular, the speed of the MAM-SBR was about three times higher than that of the MAM-CR. These results showed that the magnetic force of the magnetic microscaffold increases in proportion to its volume and magnetization, and the gradient magnetic field. This speed difference between the MAM-CR and the MAM-SBR enables the sequential delivery of the magnetic microscaffolds to the subchondral bone and cartilage.

### 3.4. Targeting Validation of Magnetic Microscaffold

Based on the mobility results of the MAM-CR and MAM-SBR, we performed targeting tests of both types of magnetic microscaffolds under the same external magnetic field. First, the sequential delivery of the MAM-CR and MAM-SBR was tested in a chamber filled with a glycerol solution under a magnetic field generated by a neodymium magnet (Figure 5a and Appendix A). Randomly mixed MAM-CR and MAM-SBR were simultaneously injected into the chamber and attracted to the target area where the magnet was located, generating a strong gradient magnetic field. Here, the MAM-SBRs with relatively strong magnetic forces reached the target before the MAM-CRs.

Next, to confirm the feasibility of using the proposed MAM-CR and MAM-SBR for osteochondral regeneration, we conducted targeting of the magnetic microscaffolds in the 3D knee joint phantom. As shown in Figure 5b, the target area was formed in the medial condyle of the femur, where cartilage defects occur frequently in the knee joint phantom, and the phantom filled with PBS was located in the workspace of the EMA system. Five MAM-SBRs and ten MAM-CRs were guided to the target area under a magnetic field of 60 mT and a gradient magnetic field of 0.9 T/m generated from the EMA system (Figure 5c and Appendix A). As a result, we confirmed that the MAM-CRs and MAM-SBRs were sequentially delivered to the target area within the phantom under the magnetic guidance of the EMA system. These results show that the two types of magnetic microscaffolds with different magnetic forces can be delivered to the damaged osteochondral site in one step under the same magnetic field.

## 4. Conclusions

Based on the mobility results of the MAM-CR and MAM-SBR, we performed targeting tests of both magnetic microscaffold types under the same external magnetic field. First, the sequential delivery of the MAM-CR and MAM-SBR was tested in a chamber filled with a glycerol solution under a magnetic field generated by a neodymium magnet. Randomly mixed MAM-CR and MAM-SBR were simultaneously injected into the chamber and attracted to the target area where the magnet was located, generating a strong gradient magnetic field. Here, the MAM-SBRs with relatively strong magnetic forces reached the target area before the MAM-CRs. We propose PLGA-based magnetic microscaffolds for osteochondral regeneration. The magnetic microscaffolds provide pore sizes suitable for bone and cartilage regeneration through w-o-w emulsion and microfluidic flow control, respectively, and can be guided to the desired locations by external magnetic fields via MNPs attached to their surfaces. The biocompatibilities of the microscaffolds were verified through viability evaluations and observations of the cells that were loaded onto the scaffolds. The magnetic actuation performances of the microscaffolds were evaluated by measuring the speed in relation to the magnetic field strength generated by the EMA system. Finally, the feasibility of the MAM-CR and MAM-SBR in knee osteochondral regeneration was demonstrated by sequential targeting experiments in the 3D knee joint phantom. Although we developed two types of microscaffolds for osteochondral regeneration, their shortcomings need to be overcome for in vivo application tests beyond the in vitro and phantom tests. Therefore, we will test the biocompatibility and biodegradability of the microscaffolds using human-derived cells in our future work. Furthermore, we will sequentially deliver cell-loaded microscaffolds for bone and cartilage regeneration to damaged osteochondral sites in vivo using external magnetic fields and verify the in vivo regeneration effects.

## 5. Patents

E.C. and G.G. are inventors on the Korean patent 10-2114300 (granted date: 18 May 2020) and the US patent 11,452,605 B2 (application date: 27 September 2022). All patent applications were submitted by the Chonnam National University and cover the microscaffold.

## Figures and Tables

**Figure 1 micromachines-14-00434-f001:**
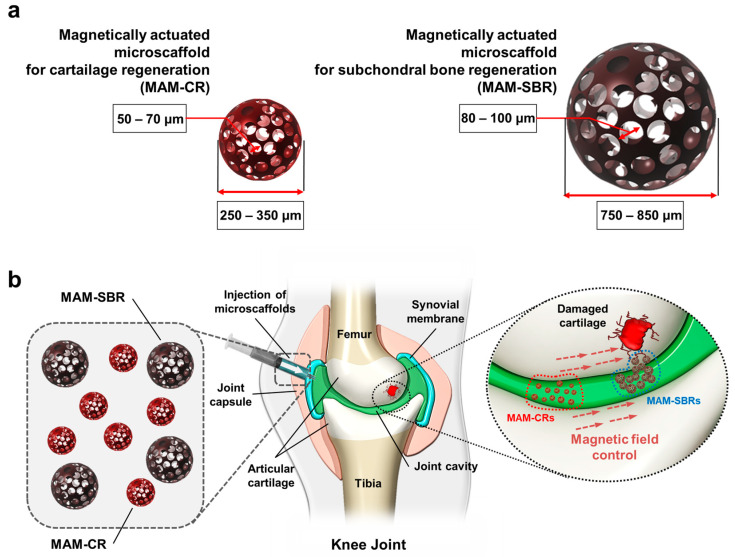
Conceptual overview of magnetically actuated microscaffolds with controllable magnetization and morphology for regeneration of osteochondral tissue. (**a**) Illustrations of magnetically actuated microscaffolds for cartilage regeneration (MAM-CR) and subchondral bone regeneration (MAM-SBR). (**b**) Targeted delivery processes of cell-loaded MAM-CR and MAM-SBR.

**Figure 2 micromachines-14-00434-f002:**
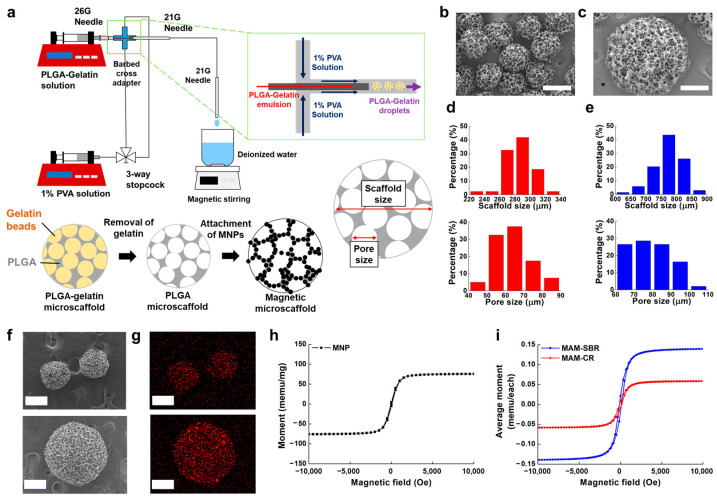
Fabrication and characterization of the magnetic microscaffolds. (**a**) Schematic illustration of the magnetic microscaffolds. SEM images of (**b**) MAM-CR and (**c**) MAM-SBR. Distributions of diameters and pore sizes of (**d**) MAM-CR (*N* = 43) and (**e**) MAM-SBR (*N* = 69). (**f**,**g**) SEM and EDX images of MAM-CR (top) and MAM-SBR (bottom). The scale bars are 250 µm. (**h**) Magnetization curves of MNPs. (**i**) Magnetization curves of MAM-CR and MAM-SBR.

**Figure 3 micromachines-14-00434-f003:**
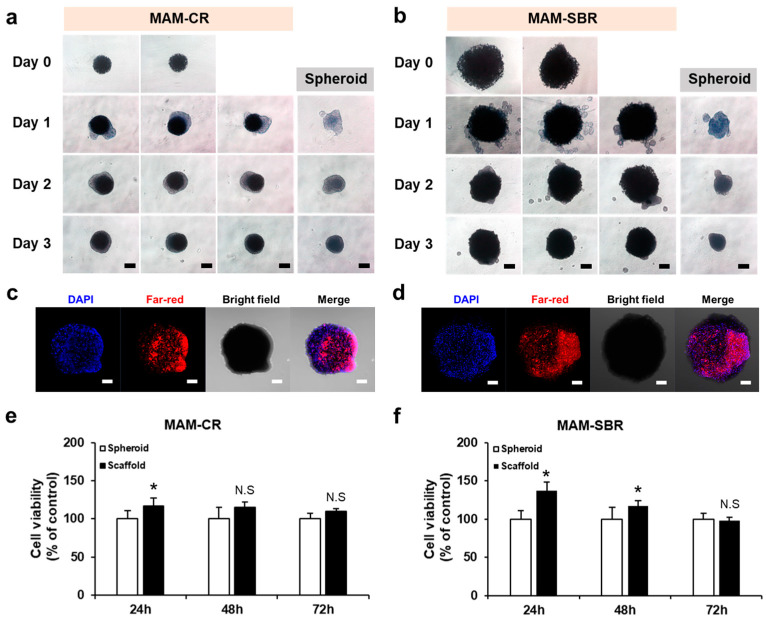
Cell loading and viability on the magnetic microscaffolds. Optical microscopy images of the cell-loaded (**a**) MAM-CR and (**b**) MAM-SBR for 3 days. The scale bars are 200 µm. Fluorescence images of the cell-loaded (**c**) MAM-CR and (**d**) MAM-SBR after 1 day of incubation. The scale bars are 100 µm. Cell viabilities of the (**e**) MAM-CR and (**f**) MAM-SBR measured for 3 days (*n* = 3, * *p* < 0.05, Student’s *t* test). N.S, not significant.

**Figure 4 micromachines-14-00434-f004:**
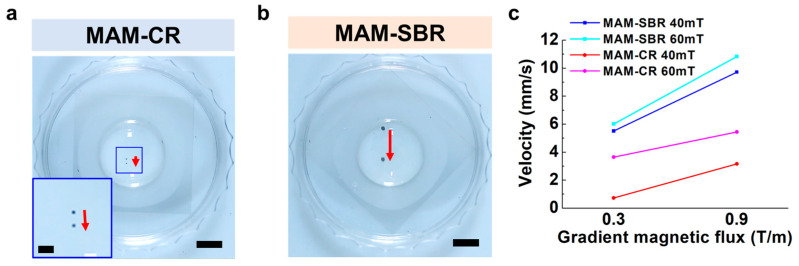
Magnetic actuation performance of the microscaffolds. Time-lapse images for the mobility of (**a**) MAM-CR and (**b**) MAM-SBR at different magnetic fields (40 mT) and gradients (0.3 T/m). The red arrows indicate the moving distances of the microscaffolds for one second. The scale bars are 5 mm. The blue boxes in (**a**) are shown as insets. The scale bar of the inset images is 1 mm. (**c**) Graph for mobilities of MAM-CR and MAM-SBR at different magnetic fields (40 and 60 mT) and gradients (0.3 and 0.9 T/m).

**Figure 5 micromachines-14-00434-f005:**
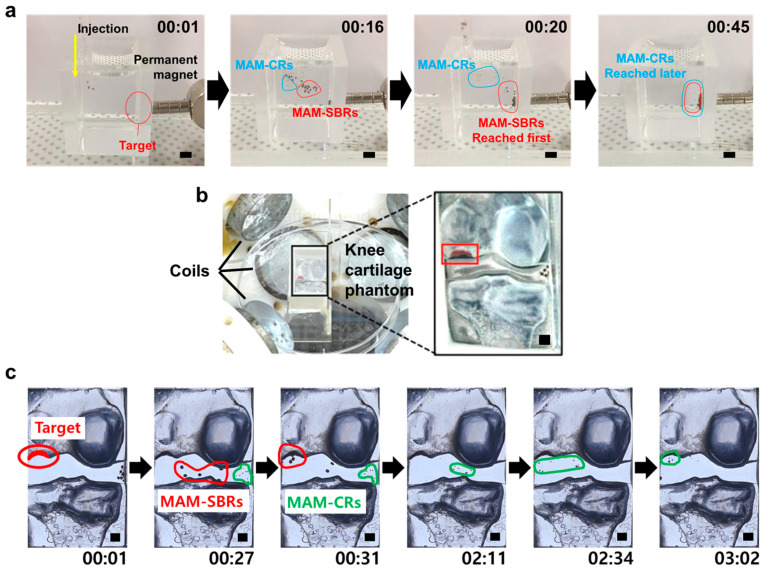
Target tests of the microscaffolds in the chamber and 3D knee joint phantom. (**a**) Time-lapse images of sequential targeting of MAM-CR and MAM-SBR in the chamber under the same magnetic field strength generated by a permanent magnet. The scale bars are 3 mm. (**b**) Experimental setup for magnetic targeting tests of the microscaffolds. The scale bars are 2 mm. (**c**) Time-lapse images of sequential targeting of MAM-CR and MAM-SBR in the 3D knee joint phantom under similar magnetic fields generated by the EMA system. The scale bars are 2 mm. The timestamp is indicated in each image in the minutes:seconds format.

## Data Availability

Not applicable.

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
