# Peer review of "Magnetically Actuated Microscaffold with Controllable Magnetization and Morphology for Regeneration of Osteochondral Tissue"

_micromachines, 2023, doi:10.3390/mi14020434_

Round 1
Reviewer 1 Report
The manuscript entitled, ‘Magnetically actuated microscaffold with controllable magneti- 2
zation and morphology for regeneration of osteochondral tissue’ reported fabrication of magnetic scaffold for osteochondral tissue enginerirng. The article nicely discussed but still I am mentioning some points which should be justified before publication;
1. First of all why magnetic scaffold is required is not clear. Better to elaborate it.
2. Magnetic sensitivity is achieved because of magnetic particles. Which one is better in this case? Ferro or paramagnetic?
3. Figure 2i looks like superparamagnetic nature. But that was not mentioned in the text. Comment on that with justification.
4. The abstract is too complex. Better to simplify it with fewer data.
5. Is there any optimum concentration of magnetic nanoparticles for optimized magnetic behavior achievement?
6. All magnetic locomotion was done is PBS. Is there any special reason? What about the choice of solvent?
7. Is the locomotion temperature sensitive? Comment on it.
8. Some articles with such topic have significance in it. Here are some for your reference: https://doi.org/10.1016/j.progpolymsci.2022.101574; https://doi.org/10.3390/polym13234259; DOI 10.1088/1361-648X/aaa344; https://doi.org/10.1016/j.jmmm.2021.168169.
Reviewer 2 Report
Thank you for being able to review the exciting work Magnetically actuated microscaffold with controllable magnetization and morphology for regeneration of osteochondral tissue.
In this work, the authors present a prototype of magnetic and microporous scaffolds with two different sizes, which allow a direction through the magnetic flux to act to work together with bone lesions and knee cartilage, thus promoting osteochondral tissue regeneration in an invasive way, and are promising in the treatment of diseases.
Overall both the idea and the manuscript are excellent, and the authors are to be congratulated for this excellent work.
Below I will make my considerations and suggestions so that they can be considered;
Abstract: Okay! Concisely summarizes the context and results obtained in work.
Introduction: It is also succinct and clear. However, I believe it would be interesting if it were possible to add some more current references to make the relevance and topicality of the topic studied very clear.
Figure 1: It's excellent, it's succinct, and represents the proposed work very well; it would even be interesting if the journal allowed using it as a graphical abstract of the work.
Materials and Methods: They are very clear and succinct.
I only have one comment regarding the description of the EDX. It would be interesting to make it clear if the microscope allows quantitative analysis or is just a semi-quantitative analysis.
I also believe it would be interesting to put the number of scaffolds counted to obtain sample sizes' mean and standard deviation.
Results and discussion:
Figure 2 is very good. However, the amount of information it has ends up overloading it. In general, the quality of the images needs to be increased, as some of the resolutions were low, which makes some analysis difficult.
In particular, I believe that the EDX spectra could be added to the supplementary material, as this would reduce the amount of information and allow the other images to be enlarged a little. If the authors do not agree with this suggestion, I would ask that the spectra be plotted on a graph with white background, or even if the image resolution is increased because it is almost ineligible.
Fig 2 f) It would also be very interesting if it could be enlarged, especially the central region, as it seems that the materials present a magnetic hysteresis that is difficult to confirm.
Some doubts that would be interesting for the authors to make clear regarding this measure are:
Do the authors mention that for this measurement, 1000 microscaffolds were used to measure the VSM, but is it not clear how the authors did this? Since the materials are a dispersion, or even in the dry case, the authors normalize to conclude that in the measurement, exactly 1000 microscaffolds were added for the VSM measurement.
The second question is that the MNP is the same for the two scaffolds, so the curves normalized by the mass would be the same, is this what was really obtained? Perhaps, to avoid generating so many doubts, it would be nice if the authors added the original curves (not calculated for each scaffold) in the supplementary material.
In figure 3 (e) and (f) Correct the spelling in these images, "shperoid" by spheroid.
In these images, there is a significant difference between the scaffolds and the spheroid used in the comparison in the first 24 h; what would that indicate?
As for Figure 4, it would perhaps be interesting to place the 4a in the supplementary material, and the (b) and (c), it would be better to choose only one of the variations (the most significant) to make a more representative image and add the others in the supplementary material because the way it is the images are very small if possible improving the resolution of the images and contrast would also be ideal to further enhance the result obtained.
Already in the sequence addressing topics related to Figure 5 are Excellent ideas from the videos. However, it would be good to check if video 2 is the same as Figure 5c because they do not seem to be exactly the same...
Finally, regarding the conclusions, I believe they are very clear.
Reviewer 3 Report
This is an interesting and important work describing the proposed and fabricated microscaffolds for osteochondral tissue regeneration. The microscaffolds were shown to grow cells without toxicity as potential cell carriers. The microscaffolds are well characterized and tested in vitro with using mouse mesenchymal stem cells and mouse C2C12 myoblast cells. The manuscript is clearly written. The conclusions made are supported by the experiments. I recommend this work for publishing. However, before a decision is made to publish, several important comments should be addressed.
Comments
1. It is well known that nano- and micromagnets generate a high-gradient magnetic field in their vicinity, with a magnetic gradient value up to 1 MT/m [https://doi.org/10.3390/cells10071734 ]. On the other hand, a high-gradient magnetic field may affect the differentiation pathways of stem cells [https://doi.org/10.1002/bies.201800017 ]. Moreover, the C2C12 myoblast cells are also affected by a gradient magnetic field [https://doi.org/10.1016/j.biomaterials.2018.02.031 ]. Thus, the manuscript would greatly benefit if the authors added a brief discussion of possibilities of the above-mentioned effects in relation to the proposed microscaffolds. For readers, such a discussion will demonstrate a wider application potential of the proposed microscaffolds in medicine.
2. Magnetic gradient magnitude in the vicinity of both microscaffolds should be estimated. I mean the gradient value at the microscaffold surface.
3. As for application potential of the microscaffolds in vivo. Are non-magnetic microscaffolds materials biodegradable? Please clarify this point.
Round 2
Reviewer 1 Report
This can be published in its present state.